# Evidence for the Link Between Non-Motor Symptoms, Kinematic Gait Parameters, and Physical Function in People with Parkinson’s Disease

**DOI:** 10.3390/bioengineering12050551

**Published:** 2025-05-21

**Authors:** Eren Timurtas, Ahmed-Abou Sharkh, Kedar K. V. Mate, Helen Dawes, Nancy E. Mayo

**Affiliations:** 1Department of Physiotherapy and Rehabilitation, Faculty of Health Sciences, Marmara University, Istanbul 34722, Turkey; eren.timurtas@marmara.edu.tr; 2School of Physical and Occupational Therapy, Faculty of Medicine, McGill University, Montreal, QC H3G 1Y5, Canada; 3Center for Outcomes Research and Evaluation (CORE), Research Institute of McGill University Health Center (MUHC), Montreal, QC H3H 2R9, Canada; 4PhysioBiometrics Inc., Montreal, QC H2V 1P4, Canada; ahmed@physiobiometrics.com (A.-A.S.); kedar.mate@mail.mcgill.ca (K.K.V.M.); h.dawes@exeter.ac.uk (H.D.); 5Faculty of Medicine and Health Sciences, McGill University, Montreal, QC H3A 0G4, Canada; 6NIHR Exeter BRC, Medical School, Exeter University, Exeter TR10 9EZ, UK; 7Division of Geriatric Medicine, Department of Medicine, McGill University, Montreal, QC H3A 0G4, Canada

**Keywords:** Parkinson’s disease, gait, kinematics, physical function, balance, walking

## Abstract

Background: Parkinson’s disease (PD) affects both motor and non-motor functions, but their interactions are understudied. This study aims to explore the relationships between non-motor and motor effects of PD, focusing on depression, fatigue, gait parameters, concentration, and physical function. Methods: This is a secondary analysis of baseline data from a randomized feasibility study using a commercially available Heel2Toe™ sensor, providing auditory feedback for gait quality. The sample included PD patients with gait impairments who walked without aids. Non-motor measures were depression, fatigue, and concentration, while motor measures included gait quality (angular velocity and variability during heel strike, push-off, foot swing) and physical function (6MWT, Mini-BESTest, Neuro-QoL). Path analysis was used to assess direct and indirect effects. Results: Among 27 participants, fatigue impacted heel strike, which affected Neuro-QoL. Mood influenced push-off and Neuro-QoL, with a direct link to 6MWT. Foot swing affected Mini-BESTest and Neuro-QoL directly. Conclusions: Non-motor PD effects directly influenced specific gait parameters and physical function indicators, highlighting potential digital biomarkers of fatigue and mood for targeted interventions.

## 1. Introduction

Parkinson’s disease (PD) causes both motor and non-motor symptoms. Visible early motor manifestations include slowness and loss of quality of movement reflected in an altered and more variable gait pattern [1,2,3]. Motor deficits affect walking, posture, balance, and functioning in everyday activities. Parkinsons impacts on movement automaticity, increasing both physical and mental effort during walking, particularly when navigating more complex environments [4,5,6], placing people with PD at high risk of falls [7,8] and diminished quality of life [9].

Rehabilitation for people with PD mainly focuses on the motor effects of PD as they affect most aspects of function [10]. Motor focused rehabilitation interventions are strongly task oriented with improvement in walking the most common target [11]. This most recent systematic review [11] also identified that walking practice was the most common modality used in rehabilitation, although research recognizes that gait impairments underpin walking [12]. Walking competency, defined as the ability to competently and safely navigate in the community [13], requires a safe effective gait pattern, balance, core and peripheral strength, movement speed and endurance, and environmental awareness [14]. Walking and balance are highly interdependent [15,16,17]. Both are required for safe and independent capacity for activities of daily living [18,19], while improving balance improves walking and walking can improve balance [20,21,22]. Focusing on training a heel-to-toe gait pattern has been recommended to improve gait in people with PD [23], which is a way of changing posture and stride length.

The non-motor effects of PD, the invisible disabilities, share a common pathological pathway [24,25], and are known to have a direct effect on activity [26] and quality of life [9,27]. Importantly there is also evidence that these non-motor symptoms are linked to temporo–spatial gait metrics [28]. Kim et al., [29] demonstrated an association between cognitive impairment and stride length. Depression in people with PD has been linked to magnitude and variability in temporo–spatial gait parameters [30,31,32]. Muscle fatigue has also been implicated in gait and falls [33], and perception of fatigue has been linked to axial/postural/gait impairment [34]. However, the effect of non-motor symptoms on kinematic gait parameters that underlie these temporo–spatial parameters [35,36] remains understudied.

This study aimed to estimate both the direct and indirect effects of non-motor symp-toms, specifically depressive symptoms, fatigue, and concentration on kinematic gait parameters and physical function. Direct effects refer to the immediate influence of these symptoms on kinematic gait parameters, while indirect effects involve pathways medi-ated by other variables, such as temporo–spatial gait metrics. A path analysis model isemployed to disentangle these relationships, quantifying the direct and indirect effects within a structural framework. This approach aligns with the hypothesis that non-motor symptoms impact physical function both directly and indirectly through kinematic gait parameters, which in turn influence indicators of physical function.

Furthermore, this study has the potential to support the qualification of specific kinematic metrics as digital clinical outcome assessments (dCOA) for PD. These metrics will be evaluated for their reliability, sensitivity, and specificity to assess PD-related changes using established psychometric validation processes, including repeatability analyses and responsiveness to clinical change. By elucidating these relationships, the study aims to inform targeted interventions to improve physical function in individuals with PD.

## 2. Materials and Methods

This study is a secondary analysis of baseline data from a randomized pilot and feasibility study. The aim of the pilot trial was to estimate the extent to which training over a three-month period with a commercially available therapeutic wearable, the Heel2ToeTM sensor, was feasible and acceptable to participants and to estimate changes in and gait pattern and walking capacity when using the sensor to provide feedback for an optimal stepping pattern [37,38,39]. The trial was prospectively registered on 3 April 2020 under the title “Improving Walking With Heel-To-Toe Device” on ClinicalTrials.gov (NCT04300348). The project was approved by the Research Ethics Board of the McGill University Health Center. All participants provided written informed consent [40].

The Heel2ToeTM sensor, a small device that clips to the side of shoe, uses inertial measurement units (IMU), gyroscope, and accelerometer, to measure angular velocity (AV) of the ankle during the gait cycle. For the trial, two groups were formed: one group trained with the Heel2Toe sensor and one group did not. For this re-analysis, the baseline data for the sample were analyzed.

### 2.1. Population

The sample comprised people with PD manifesting gait impairments and meeting the criterion that usual walking is without a walking aid [41], corresponding to the Hoehn and Yahr Scale of 2 to 3, and they were recruited from the Movement Disorders Clinics at McGill sites and the Quebec Parkinson Network. People with a Montreal Cognitive Assessment (MOCA) [42] score indicating cognitive impairment (≤25/30) were excluded. Participants were assessed during the “on” period of their medication schedule [43].

### 2.2. Measures

Balance, functional walking capacity, and self-reported physical function were one aspect of the motor effects of PD. Functional walking capacity was measured by the 6 Minute Walk Test (6MWT) [44]. Self-reported physical function was measured with the Neuro-QoL [45] comprising 8 items, measured on a 5-point ordinal scale, related to changing body position against gravity from lying, sitting, standing, or the floor (5 items), pushing with arms (1 item), walking (1 item), and running errands or shopping (1 item). Balance was measured with the Mini-BESTest (Balance Evaluation Systems Test) which comprises 14 static and dynamic balance tests performed by the participant and graded by a trained evaluator from score 0 to 2. Higher values on these three tests indicate better physical function [46].

Three constructs were used to represent the non-motor effects of PD: ratings of depression, fatigue, and concentration. Symptoms of depression and fatigue were measured using Visual Analogue Health States [47] on a 0 to 100 scale with higher values indicating poorer states. Values greater than 40 are considered to reflect a clinical situation where treatment might be indicated [48,49]. There was no specific measure of concentration but there was one item on the 8-item Parkinson Deficit Questionnaire (PDQ) [50] querying problems with concentration when reading or watching television, measured on a 5-point ordinal scale from never to always.

Other motor effects of PD were indicators of gait quality obtained directly from the IMUs in the Heel2Toe™ sensor worn without feedback during the baseline assessment of the 6MWT. These indicators are angular velocity (AV) of the ankle joint during heel strike, push-off, and foot swing and their associated coefficients of variation (CV). AV of heel strike and push-off have a negative sign and AV of foot swing has a positive sign, thus, the magnitude of the path parameters is of relevance, not the sign [51].

### 2.3. Analysis

We hypothesized that non-motor symptoms would affect gait kinematics which would affect physical function requiring mediation analysis [52]. A path analysis was conducted to estimate both direct and indirect effects [53]. The path analysis parameters included direct effects between variables, expressed as regression coefficients (ß), with associated standard errors (SE), *p*-values, and the ratio of ß/SE, which is the equivalent to a *t*-test. As each of the path variables has a different measurement scale, the regression parameters need to be standardized to compare path strengths. This process also accounts for the dual roles of some variables that are both an explanatory (left hand side) variable and an outcome (right hand side) variable. To this end the parameter StdYX is presented, and it is interpreted as the effect of a left-hand side path variable on a right-hand side path variable such that for every SD difference on a left-hand variable, the right-hand variable differs by ßSD units. As the sample size was restricted, we wanted to test a path model that was as parsimonious as possible. Correlations of all variables were carried out and variables with weak values were not included.

### 2.4. Sample Size

This analysis used an existing dataset with 27 participants. We used a sample size calculator for structural equation modeling (SEM) to identify what effect size was detectable with the sample size [54]. We considered that the multi-item Mini-BESTest and the Neuro-QoL measures as latent variables. Our model therefore comprised 2 latents and 7 observed variables. The minimum sample size for effect estimation of 0.5 or greater was 23. The sample size to test the model structure would be much greater (*n* = 138).

## 3. Results

Table 1 presents the characteristics of the 27 participants with PD (mean age 70 years; 70% men) contributing data for this analysis (Table 1). On the non-motor effects of depressed mood and fatigue, participants scored in the concerning range (>40/100), and 25.9% reported often having trouble concentrating (18.5% sometimes).

The 27 participants contributed a total of 1354 steps, with the number of steps during baseline assessment ranging from 10 to 170. AVs and AVCVs for heel strike, push-off, and foot swing were averaged across all steps taken by each participant during the baseline 6MWT assessment [55]. The mean AV during heel strike was −141.0 (SD: 96.5). During push-off it was −105.8 (SD: 66.7), and during foot swing it was 196.4 (SD: 69.3). CVs for these kinematic gait parameters were over 75% for heel strike and push-off, and 32.0% for foot swing [51,56,57].

For balance, the average score on the Mini Best test was 19.4 (SD: 6.6), whereas the age normative value is estimated at 23. The average 6MWT was 391.2 m (SD: 153.9), approximately 72% of that predicted for age. The average value on the Neuro-QoL lower extremity measure was 34.0 (SD = 4.7), whereas the optimal value is 40.

While both AVs and AVCVs for heel strike, push-off, and foot swing were potentially relevant for the path analysis, the correlations between AVCVs and the physical function outcomes were low, so we only included AVs.

The results of the path analysis are presented in Figure 1. All 27 direct and indirect paths are hypothesized. The three non-motor symptoms are on the left-hand side of the model, termed exogenous variables and are allowed to have paths to motor symptoms directly as well as through paths to kinematic parameters, which are hypothesized to have paths to motor symptoms. Figure 1 shows that fatigue was associated with heel strike directly and with Neuro-QoL indirectly through heel strike. Mood was associated with push-off and 6MWT directly and with Neuro-QoL indirectly through push-off. Direct effects were observed for heel strike and push-off with Neuro-QoL, and foot swing was associated with balance and 6MWT. Concentration was not associated with any path variable.

To complement the path parameters presented in Figure 1 showing the connections across levels, we present in Table 2 the extent to which the variables at each level are correlated (Figure 1, Table 2). At the level of non-motor symptoms, correlations were low; kinematic parameters were moderately correlated; for physical function measures, balance (Mini-BEST) and 6MWT were highly correlated, and self-reported mobility of the lower extremity (Neuro-QoL) showed low correlation with these performance measures.

Table 3 summarizes the direct path parameters (Table 3). The ones that reached significance are presented in bold font. The parameter labeled StdYX allows a comparison to be made for each pair-wise path. The three strongest direct effects were for heel strike AV and Neuro-QoL (−0.572), foot-swing AV and Mini-BEST (0.529), and mood and 6MWT (0.440). As there were indirect effects of fatigue on Neuro-QoL through heel strike AV, the total effect of fatigue on Neuro-QoL was 0.382 + |0.127|, totaling 0.509. The total effects of mood on Neuro-QoL are given by |0.232 + 0.003|, where |r| represents the absolute value of the path parameter. Since AV for heel strike and push-off are negative, the absolute value notation is used.

## 4. Discussion

Our study investigated the relationships between non-motor symptoms and the motor effects of Parkinson’s disease (PD), focusing on their effects on kinematic gait parameters and physical function [58,59]. We found that fatigue and mood significantly affected kinematic gait parameters. Fatigue was directly linked to heel strike parameters, affecting the initial phase of gait, while mood was associated with push-off parameters and the 6-Minute Walk Test (6MWT), influencing the terminal stance phase and walking endurance. These findings are interesting as parameters relating to fatigue may predispose to less stable gait, and factors associated with mood may reduce walking propulsion. Non-motor symptoms also exert indirect effects on physical function through kinematic gait parameters. Fatigue indirectly affected Neuro-QoL via heel strike, suggesting that addressing fatigue could improve gait kinematics and physical function. Similarly, the indirect effect of mood on Neuro-QoL through push-off highlighted the connection between emotional well-being and motor function. Kinematic parameters had direct effects on physical function measures. Heel strike and push-off were directly linked to Neuro-QoL, underscoring their importance in functional mobility. Foot swing was associated with balance (Mini-BEST) and 6MWT, emphasizing its role in stability and endurance. Significant direct path parameters included heel strike AV on Neuro-QoL (−0.572), foot swing AV on Mini-BEST (0.529), and mood on 6MWT (0.440). The total effects showed the impact of non-motor symptoms on physical function, with the total effect of fatigue on Neuro-QoL being 0.509 and mood on Neuro-QoL being 0.235.

### 4.1. Fatigue and Its Impact on Gait and Physical Function

Our results indicate that fatigue significantly affects motor symptoms and physical function, particularly impacting heel strike parameters. This aligns with the study by Hagell et al. [34], which found fatigue to be a prominent symptom in PD, associated with increased Hoehn and Yahr stages and various non-motor symptoms like anxiety, depression, and lack of motivation. The association of fatigue with axial/postural/gait impairment but not with other motor symptoms such as tremor or rigidity further supports our findings that fatigue influences specific aspects of gait. In the review by Ghani et al. [33], lower limb muscle fatigue was shown to alter gait performance, increasing stride length and reducing stride duration, which could potentially increase the risk of falls. This supports our findings that fatigue impacts the initial phase of gait (heel strike), suggesting that interventions to reduce fatigue could improve gait stability and reduce fall risk. The work by Terra et al. [60], found fatigue did not have a direct relationship with balance but they did not investigate indirect effects.

### 4.2. Mood and Emotional Well-Being and Their Impact on Gait and Physical Function

Mood significantly affected push-off parameters and the 6-Minute Walk Test (6MWT, a measure of walking endurance. This is consistent with the findings by Kincses et al. [30], who reported that depressive symptoms in PD patients were associated with altered gait characteristics, including reduced velocity and shorter stride length, which in turn affected health-related quality of life. Our study further demonstrates that mood has indirect effects on physical function through its impact on gait parameters, underscoring the interconnectedness of emotional well-being and motor function. The influence of mood on push-off and 6MWT is consistent with research by Liguori et al. [28], who found moderate correlations between non-motor symptoms like pain and fatigue with motor performance in PD—specifically gait duration and values on the Timed-Up-and-Go test (TUG) under single and dual tasks conditions.

### 4.3. Indirect Effects of Non-Motor Symptoms on Physical Function

Our path analysis revealed that non-motor symptoms exert indirect effects on physical function mediated through kinematic gait parameters. For instance, fatigue had an indirect effect on Neuro-QoL via heel strike, suggesting that addressing fatigue could enhance gait kinematics and quality of life. This mediation pathway was also noted in the study by Kincses et al. [30], which highlighted the association between gait characteristics and depression in PD patients, reinforcing the importance of considering emotional health in gait and physical function assessments [60].

### 4.4. Kinematic Parameters and Effects on Physical Function

Kinematic parameters, including heel strike and push-off, directly influence various aspects of physical function such as Neuro-QoL, Mini-BEST, and 6MWT. Significant direct path parameters were heel strike AV on Neuro-QoL (−0.572), foot swing AV on Mini-BEST (0.529), and mood on 6MWT (0.440). This is corroborated by the research of Lord et al. [32,61], who found that gait speed and step variability were crucial determinants of balance and functional mobility in PD patients. Our findings emphasize that the precise timing and force of these gait phases are critical for maintaining physical function.

### 4.5. Strength and Limitations

A major strength of this study is the use of path analysis to explore the direct and indirect relationships between non-motor symptoms, kinematic gait parameters, and physical function. This methodological approach allowed us to elucidate complex interactions and mediation effects, providing a nuanced understanding of how non-motor symptoms impact physical function in PD. Our study emphasizes a multi-factorial perspective, considering both motor and non-motor symptoms. This holistic approach highlights the importance of addressing the full spectrum of PD symptoms, which is crucial for developing more effective and comprehensive treatment strategies.

A notable limitation of this study is the relatively small sample size. While the results provide valuable insights, the generalizability of our findings may be limited. We also had a limited portfolio of measures which was necessary to reduce the respondent burden so that the focus could be on the processes of the trial. Thus, our measure of cognition was restricted to a single item for concentration. The observation that concentration did not have any impact may be due to under-specification of this important construct. A richer set of measures for mood and fatigue would have been optimal allowing for a structural equation model to be fit to the data and permit a focus to be on constructs rather than individual measures, which less accurately reflect the constructs of interest.

The cross-sectional nature of the study limits our ability to infer causality. Future studies with larger cohorts are necessary to confirm these relationships and ensure the robustness of the conclusions drawn.

The potential of wearable technology to transform PD management is promising but faces challenges in under-resourced areas, including high costs, limited access, and insufficient training. Future research should focus on subsidizing costs, integrating wearables into existing systems, and designing affordable, user-friendly devices. Collaborations with policymakers and industry stakeholders will be key to ensuring equitable access and scalability.

## 5. Conclusions

The findings of this study provide important insights for the current management of PD. The clear relationships identified between non-motor symptoms, kinematic gait parameters, and physical function suggest that a comprehensive approach to treatment is essential. Clinicians should consider not only the direct motor symptoms but also the significant impact of non-motor symptoms such as fatigue and mood on patients’ gait and overall physical function. This implies that current therapeutic strategies should be broadened to include interventions targeting non-motor symptoms, gait mechanics, as well as physical function. For now, these results advocate for the integration of multi-disciplinary teams in the care of PD patients. Neurologists, physical therapists, and mental health professionals need to work collaboratively to address both motor and non-motor symptoms. For example, targeted therapies to alleviate fatigue could indirectly benefit gait parameters, while interventions aimed at improving mood might enhance endurance and physical function. The incorporation of these comprehensive treatment plans can lead to more effective management of PD, reducing the burden of the disease on patients.

This study lays the groundwork for further exploration into the complex interplay between non-motor and motor symptoms in PD. Future research should investigate the mechanisms by which non-motor symptoms influence gait kinematics and physical function. Longitudinal studies could provide more definitive evidence on the causal relationships and the long-term effects of integrated treatment approaches.

The advancement of wearable technology and mobile health applications presents an exciting opportunity for monitoring and managing PD motor effects in real-time. Future research could explore how these technologies can be used to track gait parameters, providing personalized feedback, and enabling timely interventions. The development of such tools could revolutionize the management of PD, making it possible to tailor treatments to individual patients’ needs more precisely. The findings highlight the potential of personalized medicine in treating PD. By recognizing the unique constellation of symptoms each patient experiences, healthcare providers can develop more targeted and effective treatment plans. Future studies should focus on identifying biomarkers that predict individual responses to different therapies, facilitating a more personalized approach to managing both motor and non-motor symptoms. On a broader scale, these findings should inform healthcare policies aimed at improving the standard of care for PD patients. Policies that encourage the integration of mental health support into PD management plans and the funding of multi-disciplinary care teams could enhance patient outcomes. Additionally, investment in research and development of new technologies for monitoring and treating PD will be crucial in addressing the growing prevalence of this condition.

In conclusion, the insights gained from this study highlight the potential of using gait parameters as dCOAs for non-motor symptoms and suggest areas for intervention.

## Figures and Tables

**Figure 1 bioengineering-12-00551-f001:**
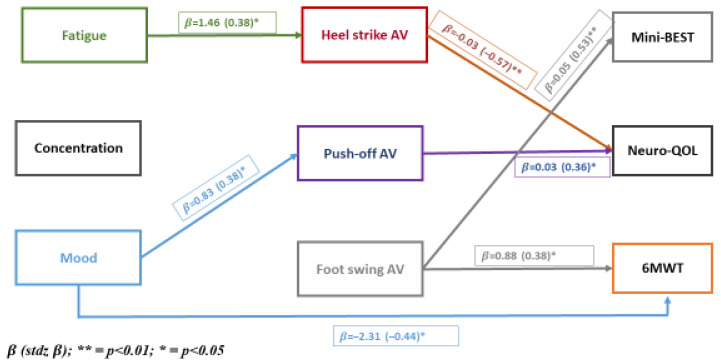
Path model.

**Table 1 bioengineering-12-00551-t001:** Characteristics of participants (*n* = 27).

Variables	Mean ± SD or *n* (%)
Age (years)	70.3 ± 8.4
Min-Max	51–84
Women/Men	8/19 [30/70]
Non-motor symptoms (higher is worse)	
Depressed mood (0–100)	41.1 ± 30.7
Fatigue (0–100)	47.2 ± 25.0
Trouble Concentration	17.7 ± 4.1
Never	0 (0)
Occasionally	14 (51.9)
Sometimes	5 (18.5)
Often	7 (25.9)
Kinematic gait parameters (angular velocity: °/sec)	
Heel strike	−141.0 ± 96.5
Push-off	−105.8 ± 66.7
Foot swing	196.4 ± 69.3
Kinematic gait parameters (coefficient of variation: %)	
Heel strike	76.3%
Push-off	77.9%
Foot swing	32.0%
Physical function (higher is better)	
Mini-BESTest (0–28) (age norm 23)	19.4 ± 6.6
6MWT (m) (age norm 545)	391.2 ± 153.9
Neuro-QoL (8–40)	34.0 ± 4.7

**Table 2 bioengineering-12-00551-t002:** Spearman correlations among path variables at the same level.

		*n*	r	95% CI
Mood	Fatigue	25	0.213	[−0.20. 0.56]
Mood	Concentration	26	0.178	[−0.22. 0.53]
Fatigue	Concentration	27	−0.148	[−0.50. 0.25]
Heel strike AV	Push-off AV	27	0.638	[0.34. 0.82]
Heel strike AV	Foot swing AV	27	−0.562	[−0.78. −0.23]
Push-off AV	Foot swing AV	27	−0.439	[−0.70. −0.07]
Mini-BESTest	6MWT	26	0.723	[0.47. 0.87]
Mini-BESTest	Neuro-QoL	26	0.290	[−0.11. 0.61]
6MWT	Neuro-QoL	25	0.194	[−0.22. 0.55]

AV: Angular Velocity.

**Table 3 bioengineering-12-00551-t003:** Direct path parameters for the model.

Explanatory	Outcome	*β*	SE	*p*	t-Value (*β*/*se*)	StdYX
**Fatigue**	**Heel strike AV**	**1.459**	**0.679**	**0.032**	**2.149**	**0.382**
Fatigue	Push off AV	0.423	0.480	0.378	0.881	0.158
Fatigue	Foot swing AV	−0.981	0.519	0.059	−1.890	−0.353
Fatigue	Mini-BESTest	0.058	0.053	0.271	1.100	0.216
Fatigue	6MWT	2.292	1.252	0.067	1.831	0.357
Fatigue	Neuro-QoL	0.001	0.039	0.993	0.009	0.002
Mood	Heel strike AV	0.809	0.555	0.145	1.456	0.259
**Mood**	**Push off**	**0.833**	**0.391**	**0.033**	**2.132**	**0.381**
Mood	Foot swing AV	−0.146	0.425	0.731	−0.343	−0.064
Mood	Mini-BESTest	−0.062	0.042	0.138	−1.483	−0.280
**Mood**	**6MWT**	**−2.308**	**0.984**	**0.019**	**−2.346**	**−0.440**
Mood	Neuro-QoL	−0.042	0.030	0.165	−1.389	−0.232
Concentration	Heel strike AV	−7.349	19.190	0.702	−0.383	−0.068
Concentration	Push off AV	−14.601	13.515	0.280	−1.080	−0.192
Concentration	Foot swing AV	6.160	14.667	0.675	.420	0.078
Concentration	Mini-BESTest	−0.585	1.307	0.655	−0.447	−0.076
Concentration	6MWT	5.143	30.962	0.868	0.166	0.028
Concentration	Neuro-QoL	−0.725	0.947	0.444	−0.766	−0.115
Heel strike AV	Mini-BESTest	0.005	0.013	0.730	0.345	0.064
Heel strike AV	6MWT	−0.017	0.311	0.955	−0.056	−0.010
**Heel strike AV**	**Neuro-QoL**	**−0.033**	**0.010**	**<0.01**	**−3.452**	**−0.572**
Push off AV	Mini-BESTest	−0.001	0.019	0.954	−0.058	−0.011
Push off AV	6MWT	0.189	0.441	0.668	0.429	0.079
**Push off AV**	**Neuro-QoL**	**0.030**	**0.014**	**0.030**	**2.175**	**0.358**
**Foot swing AV**	Mini-BESTest	**0.051**	**0.017**	**0.003**	**3.010**	**0.529**
**Foot swing AV**	**6MWT**	**0.884**	**0.406**	**0.029**	**2.180**	**0.383**
Foot swing AV	Neuro-QoL	−0.011	0.013	0.394	−0.853	−0.134

SE: standard error.

## Data Availability

Data could be made available for inclusion in meta-analysis.

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
