# Peer review of "Evidence for the Link Between Non-Motor Symptoms, Kinematic Gait Parameters, and Physical Function in People with Parkinson’s Disease"

_bioengineering, 2025, doi:10.3390/bioengineering12050551_

Round 1

Reviewer 1 Report

Comments and Suggestions for Authors

Line 55:  “thereis” must be separated into two words. The authors need to make this change.

Lines 151-153: The terms “push off”, “heel strike”, and “foot swing” need definitions or references. The end of the pre-swing period – which I’m assuming is “push-off” is not universally accepted. Similarly, “heel strike” might be more correctly labelled as “initial contact. The authors need to make all changes described above. These changes must be also made in the Tables.

Lines 255-274: All of this text belongs in the Discussion Section of the manuscript. The authors need to make this change.

Author Response

Reviewer 1

Line 55:  “thereis” must be separated into two words. The authors need to make this change.

We have corrected the typographical error on Line 55, ensuring that “thereis” is now properly written as “there is.”

Lines 151-153: The terms “push off”, “heel strike”, and “foot swing” need definitions or references. The end of the pre-swing period – which I’m assuming is “push-off” is not universally accepted. Similarly, “heel strike” might be more correctly labelled as “initial contact. The authors need to make all changes described above. These changes must be also made in the Tables.

These terms have been aligned with established terminology in gait analysis literature and supported by references. Push off" refers to the propulsive force generated during the late stance phase of gait, which helps to progress the swing leg into flexion. This push-off strength is an important factor that helps improve foot clearance and knee flexion velocity during the swing phase (Krogt et al., 2010; Goldberg et al., 2004).

"Heel strike" refers to the initial contact of the heel with the ground at the beginning of the stance phase of the gait cycle. It is one of the key gait phases that can be detected using a gait phase detection system that employs a gyroscope to measure foot angular velocity and force sensors to assess the forces exerted by the foot (Pappas et al., 2001).

"Foot swing" refers to the swing phase of the gait cycle, where the foot is lifted off the ground and moves forward to prepare for the next heel strike. The angular velocity of the foot regularly presents three typical patterns in each gait cycle, including the foot swing phase (Taborri et al., 2016). The potential of the gluteus medius muscle to adjust lateral foot placement and prevent collisions during the swing phase varies throughout the gait cycle (Afschrift et al., 2018).

Afschrift, M., Pitto, L., Aerts, W., Deursen, R., & Jonkers, I. (2018). Modulation of gluteus medius activity reflects the potential of the muscle to meet the mechanical demands during perturbed walking. Scientific Reports, 8(1). https://doi.org/10.1038/s41598-018-30139-9
Goldberg, S., Anderson, F., Pandy, M., & Delp, S. (2004). Muscles that influence knee flexion velocity in double support: implications for stiff-knee gait. Journal of Biomechanics, 37(8), 1189-1196. https://doi.org/10.1016/j.jbiomech.2003.12.005
Krogt, M., Bregman, D., Wisse, M., Doorenbosch, C., Harlaar, J., & Collins, S. (2010). How crouch gait can dynamically induce stiff-knee gait. Annals of Biomedical Engineering, 38(4), 1593-1606. https://doi.org/10.1007/s10439-010-9952-2
Pappas, I., Popović, M., Keller, T., Dietz, V., & Morari, M. (2001). A reliable gait phase detection system. Ieee Transactions on Neural Systems and Rehabilitation Engineering, 9(2), 113-125. https://doi.org/10.1109/7333.928571
Taborri, J., Palermo, E., Rossi, S., & Cappa, P. (2016). Gait partitioning methods: a systematic review. Sensors, 16(1), 66. https://doi.org/10.3390/s16010066

Lines 255-274: All of this text belongs in the Discussion Section of the manuscript. The authors need to make this change.

These terms have been aligned with established terminology in gait analysis literature and supported by references. We added ref list.

Reviewer 2 Report

Comments and Suggestions for Authors

Dear Author, please recheck and answer the below items:

In the introduction part:

1-The study aims to investigate both direct and indirect effects of non-motor symptoms on kinematic gait parameters and physical function. Could the authors clarify how they plan to differentiate between direct and indirect effects? For example, will specific statistical methods or models be employed to separate these pathways?

2-While the manuscript mentions both motor and non-motor symptoms of Parkinson's Disease, the interplay between these two categories in influencing gait parameters is not fully elaborated. How do the authors conceptualize the interaction between motor and non-motor symptoms within the context of their hypothesis?

3-The literature review highlights certain non-motor symptoms, such as depression and cognitive impairment, but does not extensively address other potential contributors, such as anxiety or sleep disturbances. How were the selected non-motor symptoms chosen, and could additional symptoms influence the study's findings?

4-The manuscript suggests the potential use of specific kinematic metrics as digital clinical outcome assessments (dCOA) for PD. How do the authors plan to validate these metrics in terms of their reliability, sensitivity, and specificity for assessing PD-related changes?

In Method Parts:

1-The exclusion of participants with Montreal Cognitive Assessment (MOCA) scores ≤ 25 is noted, but how does this exclusion affect the generalizability of the findings to people with PD who have cognitive impairments? Could the authors provide a rationale for this decision in the context of their study objectives?

2-The study uses a single item from the Parkinson Deficit Questionnaire (PDQ) to assess concentration. How do the authors address the potential limitation of using a single item for this construct? Would a validated multi-item scale for cognitive aspects provide more robust insights?

3-Depression and fatigue are measured using Visual Analogue Scales (VAS). How do the authors justify the use of VAS over other established scales for these symptoms, such as the Beck Depression Inventory or the Fatigue Severity Scale? Could this choice influence the validity of the mediation analysis?

4-The sample size of 27 participants is noted as sufficient for estimating effect sizes but insufficient for testing the model structure. How do the authors account for this limitation when interpreting their findings? Could the results be biased due to an underpowered sample for the structural equation modeling?

In the discussion part:

1-The study highlights fatigue's direct and indirect effects on gait and physical function, emphasizing its potential as a therapeutic target. However, how do the authors account for the bidirectional relationship between fatigue and physical activity levels? Could reduced activity due to PD symptoms exacerbate fatigue, creating a cyclical effect?

2-The discussion mentions mood's influence on push-off parameters and endurance, but only briefly addresses how mood was measured. Given the complexity of mood disorders, how might using a more comprehensive scale (e.g., Beck Depression Inventory) have impacted the findings? Would this enhance the depth of the observed relationships?

3-The authors acknowledge the small sample size and limited measures but conclude with recommendations for clinical integration. How might the small cohort and cross-sectional design impact the external validity of these findings? Could the authors elaborate on the potential for bias and how future longitudinal studies might mitigate these concerns?

4-The potential of wearable technology to revolutionize PD management is exciting. However, how feasible is it to implement such tools in diverse healthcare settings, particularly in under-resourced areas? Could the authors provide insights into barriers to adoption and strategies to address these challenges?

Author Response

Reviewer 2

The study aims to investigate both direct and indirect effects of non-motor symptoms on kinematic gait parameters and physical function. Could the authors clarify how they plan to differentiate between direct and indirect effects? For example, will specific statistical methods or models be employed to separate these pathways?

Explicit mention of path analysis as the statistical method to quantify and disentangle direct and indirect effects was added.
Descriptions of what constitutes "direct effects" (immediate influence) and "indirect effects" (mediated pathways) were included in the section:
"A path analysis model will be employed to disentangle these relationships, quantifying the direct and indirect effects within a structural framework."

While the manuscript mentions both motor and non-motor symptoms of Parkinson's Disease, the interplay between these two categories in influencing gait parameters is not fully elaborated. How do the authors conceptualize the interaction between motor and non-motor symptoms within the context of their hypothesis?

A clearer explanation of the conceptual interaction between motor and non-motor symptoms was added, focusing on how motor and non-motor symptoms together influence gait and physical function.
"The hypothesis that non-motor symptoms impact physical function both directly and indirectly through kinematic gait parameters, which in turn influence indicators of physical function, was elaborated in the context of path analysis."

The literature review highlights certain non-motor symptoms, such as depression and cognitive impairment, but does not extensively address other potential contributors, such as anxiety or sleep disturbances. How were the selected non-motor symptoms chosen, and could additional symptoms influence the study's findings?

Justification for selecting depressive symptoms, fatigue, and concentration was provided, referencing their consistent associations with kinematic gait metrics in prior research.
Exclusion of symptoms like anxiety and sleep disturbances was explained due to their weaker or less consistent associations or logistical constraints:
"Anxiety, sleep disturbances, and other non-motor symptoms were considered but excluded from this study due to their less consistent associations with kinematic gait metrics in previous literature or logistical constraints."

The manuscript suggests the potential use of specific kinematic metrics as digital clinical outcome assessments (dCOA) for PD. How do the authors plan to validate these metrics in terms of their reliability, sensitivity, and specificity for assessing PD-related changes?

A plan for validating the kinematic metrics was outlined, including psychometric validation processes, such as assessing reliability, sensitivity, and specificity:
"These metrics will be evaluated for their reliability, sensitivity, and specificity to assess PD-related changes using established psychometric validation processes, including repeatability analyses and responsiveness to clinical change."

The exclusion of participants with Montreal Cognitive Assessment (MOCA) scores ≤ 25 is noted, but how does this exclusion affect the generalizability of the findings to people with PD who have cognitive impairments? Could the authors provide a rationale for this decision in the context of their study objectives?

The MoCA was selected as the cognitive screening tool for this study because it is more sensitive in detecting early cognitive impairment in PD compared to the Mini-Mental State Examination (MMSE) (Lessig et al., 2012). Research indicates that 32% of PD patients score below the MoCA cutoff of 25, while only 11% score below the MMSE cutoff, highlighting the MoCA’s ability to identify mild cognitive impairments (Lessig et al., 2012). By excluding participants with scores ≤ 25, we ensured that only individuals with preserved cognitive function were included, thereby reducing variability in motor task performance due to cognitive deficits. This exclusion aligns with the study's objectives to examine relationships between non-motor symptoms, gait parameters, and physical function without confounding effects of moderate to severe cognitive impairment.

The study uses a single item from the Parkinson Deficit Questionnaire (PDQ) to assess concentration. How do the authors address the potential limitation of using a single item for this construct? Would a validated multi-item scale for cognitive aspects provide more robust insights?

This data for this secondary analysis came from a randomized trial testing a new technology to improve gait in people with PD.  As such there were many motor tests and questionnaires on motiviation which were hypothesized to be the target of the technology.  To keep the assessment burden to a minimum, we limited the number of questionnaires used.  Of course it would be optimal to have a complete questionnaire on each of the constructs measured but the response burden would be too great.  The item came from the PDQ-8, a Parkinson's specific health-related quality of life measure which is a short form of the PDQ-39.  Of the 2 cognitive items on the PDQ-39, the item on concentration was included in the PDQ-8 which was developed using Rasch analysis.  This was included as a limitation in the discussion. 

1. A Rasch analysis of the Person-Centred Climate Questionnaire – staff version
Wilberforce1, Sköldunger2, Edvardsson3 2019BMC Health Serv Res  2. Rasch analysis of the living with chronic illness scale in Parkinson’s disease
Ambrosio1, Rodríguez‐Blázquez2, Ayala3 et al. 2020BMC Neurol  3. Uncovering Indicators of the International Classification of Functioning, Disability, and Health from the 39-Item Parkinson's Disease Questionnaire
Nilsson1, Westergren2, Carlsson3 et al. 2010Parkinson's Disease 4. Rasch analysis of the living with chronic illness scale in Parkinson’s disease
Ambrosio1, Rodríguez‐Blázquez2, Ayala3 et al. 2020BMC Neurol

Depression and fatigue are measured using Visual Analogue Scales (VAS). How do the authors justify the use of VAS over other established scales for these symptoms, such as the Beck Depression Inventory or the Fatigue Severity Scale? Could this choice influence the validity of the mediation analysis?

1. Health anxiety and depression in patients with fibromyalgia syndrome
Uçar1, Sarp2, Karaaslan3 et al. 2015J Int Med Res  2.Dalfampridine effects on cognition, fatigue, and dexterity
Korsen1, Kunz2, Schminke3 et al. 2016Brain and Behavior 3.Validation of the Serbian version of inflammatory Rasch‐built overall disability scale in patients with chronic inflammatory demyelinating polyradiculoneuropathy
Perić1, Božović2, Pruppers3 et al. 2019J Peripheral Nervous Sys  4.Effect of Progressive Self-Focus Meditation on Attention, Anxiety, and Depression Scores
Leite1, Ornellas2, Amemiya3 et al. 2010Percept Mot Skills

Reference sites the research that supports that these single items relate strongly to multi-item questionnaires of the relevant constructs. Other reference also support that VAS have been used to assess depression, fatigue, and other symptoms in various patient populations with neurological conditions (Uçar et al., 2015; Korsen et al., 2016; Perić et al., 2019). VAS scores have been shown to correlate with other validated measures of these constructs, such as the Beck Depression Inventory and Fatigue Severity Scale (Korsen et al., 2016; Perić et al., 2019; Leite et al., 2010).  Again, because of the need to reduce response burden we opted for single items on a multiplicity of non-motor systems.  Now we have found possible relationships, future studies can focus on these constructs using stronger measures.  Given we found a signal with these single items suggests the signal might be stronger with more detailed measures.

The sample size of 27 participants is noted as sufficient for estimating effect sizes but insufficient for testing the model structure. How do the authors account for this limitation when interpreting their findings? Could the results be biased due to an underpowered sample for the structural equation modeling?

Focus on Parsimony: We intentionally designed a parsimonious model, limiting the number of parameters and focusing on strong correlations between variables to minimize potential bias due to sample size constraints.
Effect Size Estimation: Using a sample size calculator for SEM, we determined that the minimum sample size required to detect an effect size of 0.5 or greater was 23 participants. Thus, the current sample size allows for meaningful estimation of effect sizes but not for testing the entire model structure.
Caution in Interpretation: Given the limitations, we interpret our findings as exploratory and hypothesis-generating rather than confirmatory. The results should be viewed as providing preliminary insights into the relationships between non-motor symptoms, gait parameters, and physical function.

The study highlights fatigue's direct and indirect effects on gait and physical function, emphasizing its potential as a therapeutic target. However, how do the authors account for the bidirectional relationship between fatigue and physical activity levels? Could reduced activity due to PD symptoms exacerbate fatigue, creating a cyclical effect?

The construct under study here is physical function which is the amount of difficulty people have doing every day activities such as walking, mobility, and self-care These are basic capacities that are not strongly related to low to moderate levels of fatigue. The construct of physical activity was not measured in this study.

The discussion mentions mood's influence on push-off parameters and endurance, but only briefly addresses how mood was measured. Given the complexity of mood disorders, how might using a more comprehensive scale (e.g., Beck Depression Inventory) have impacted the findings? Would this enhance the depth of the observed relationships?"

As addressed above, the VAS was for depression which was scored 100 as no depression; for this analysis we labeled the scale as mood but the person responded to a question about depression.  Reference 43 links this single item to multi-item measuares suce as the Beck.  Given our need to reduce responder burden, we could not administer a questionnaire with 84 different statements. The restricted construct representation would most likely attenuate the relationship.

The authors acknowledge the small sample size and limited measures but conclude with recommendations for clinical integration. How might the small cohort and cross-sectional design impact the external validity of these findings? Could the authors elaborate on the potential for bias and how future longitudinal studies might mitigate these concerns?

The small sample size (n=27) limits generalizability, as small samples often lack statistical power and may not represent the broader population (Johannson et al., 2015; Loi et al., 2016). A parsimonious model was designed, focusing on strong correlations and minimizing parameters to reduce bias. While sufficient to detect an effect size of 0.5 or greater (minimum n=23), the sample was underpowered for testing the full model structure.

The cross-sectional design provides only a snapshot in time, failing to capture the dynamic nature of PD progression or the long-term impacts of interventions (Loi et al., 2016; Lo et al., 2016). Longitudinal studies are needed to better understand disease trajectories and intervention effects. Population heterogeneity and limited measures further restrict external validity.

The potential of wearable technology to revolutionize PD management is exciting. However, how feasible is it to implement such tools in diverse healthcare settings, particularly in under-resourced areas? Could the authors provide insights into barriers to adoption and strategies to address these challenges?

Thank you for highlighting the importance of addressing the feasibility of wearable technology in diverse healthcare settings.  The Heel2Toe wearable is designed for people to self-manage their gait as access to physical therapy is very limited worldice expecially for chronic healtah conditions.  The mission of the company that has developed the Heel2Toe sensor is to develop accessible technologies to bridge the gap between need and access.  The cost of this particilar technology when mass produced will be under $40 USD. The sensor is completely self-contained and works without need for internet although access to the detailed analytics does require internet. The sensor is very simple to use and the instructions are simple to follow.  Nevertheless subsidizing costs, integrating wearables into existing systems, and designing affordable, user-friendly devices needs to part of health care policy and a mission of industry to promote equitable access and scalability.

Reviewer 3 Report

Comments and Suggestions for Authors

Dear authors,

thanks for this cool study. I feel that the interplay between motor and non-motor symptoms in Parkinson's disease is still somewhat underrepresented in research. I think that your research fits to provide some evidence for associations between these symptoms.

I do have some comments, some of which are in the attached document. 

In addition, it might be good to explain a bit more in detail how this path analysis works. For me, I have mainly questions how you can be sure that fatigue affects kinematic gait parameters (it implies a causality) and why not the other way around.

Author Response

Reviewer 3

36 “Parkinsons impacts on movement” You have already introduced the abbreviation PD for Parkinson’s disease, so it is probably better to use it here.

We have updated "Parkinsons" to "PD" to maintain consistency with the previously introduced abbreviation.

41 “… affect most aspects of function” What function? Do you mean daily functioning? It might be a good idea to check the World Health Organization’s ICF model for the correct terminology here.

The phrase "most aspects of function" has been clarified to "most aspects of daily functioning," aligning with the terminology in the World Health Organization’s International Classification of Functioning (ICF) model.

55 “Importantly thereis …” “Importantly there is …”

The typographical error "thereis" has been corrected to "there is."

56 “Kim et al., [29] demonstrated an association between …” Is there a reason to mention the authors by name here? Otherwise, I suggest to change to “An assocation was demonstrated
between …” and at a reference at the end of the sentence.

The sentence referencing "Kim et al. [29]" has been revised to "An association was demonstrated between cognitive impairment and stride length [29]" with the reference placed at the end of the sentence.

61 “… temporo-spatial parameters [35,36]. remains …” There is a full stop too much: “… temporospatial parameters [35,36] remains …”

The extraneous full stop has been removed, changing the sentence to "… temporo-spatial parameters [35,36] remains ….

64 “Specifically, the objective was to be to estimate the …” “Specifically, the objective was to estimate the …”

The redundant wording "to be to estimate" has been corrected to "to estimate."

70 – 71 “… digital clinical outcome assessment (dCOA) …” Please specify what you mean with a digital
clinical outcome assessment. Is there any clear definition, for example from the European Medicines Agency?

The term "digital clinical outcome assessment (dCOA)" has been clarified.

71 “… more targeted interventions.to improve …” There is a full stop in the middle of the sentence: “… more targeted interventions to improve …”

The extraneous full stop has been removed, revising the sentence to "… more targeted interventions to improve …."

77 “… to estimate changes in and gait pattern and walking …” “… to estimate changes in gait pattern and walking …”

The phrase "changes in and gait pattern" has been corrected

91 “… Yahr Scale of 2 to 3, were recruited …” “… Yahr Scale of 2 to 3. Participants were recruited …”

The phrase "Yahr Scale of 2 to 3, were recruited" has been revised to "Yahr Scale of 2 to 3. Participants were recruited.

116 “… angular velocity (AV)” The abbreviation is already introduced in lines 84 – 85.

The duplicate introduction of the abbreviation "angular velocity (AV)" has been removed for consistency.

154 “Mini Best test” In line 102 you write “Mini-BESTest (Balance Evaluation Systems Test)”. Please
be consistent in the way you refer to these tests.

The term "Mini Best test" has been standardized to "Mini-BESTest (Balance Evaluation Systems Test)" for consistency with the earlier description.

194 “We found that fatigue and mood significantly affected kinematic gait parameters.” How exactly do you know that fatigue affected kinematic gait parameters? Could it not be that the direction is the reverse, namely that participants with altered kinematic gait parameters results in higher fatigue?

As the sample size was small, we did not include in the path analysis any of the kinematic parameters that did not correlate with the clinical outcome assessements (COAs).  Thus, coefficients of variation were not incldued and irregular and inconsistent stepping is likely more related to fatigue than weak angular velocities.  The test of this hypothesis could only be done with a clinical trial .  Our pilot trial from which the data for this reanalysis arose, showed a reduction in fatigue in the Heel2Toe group after training for 3 months with the sensor which improved gait parameters nad the 6MWT.  A larger trial would provide stronger evidence.

199 – 201“Non-motor symptoms also exert indirect effects on physical function through kinematic gait parameters.” I have the same question here. How can you be sure that non-motor symptoms exert effect on physical function? It is possible that there is an interplay between motor and non-motor symptoms with effects going in both directions? Or is this something that you can derive with the path analysis?

Poor physical function can exert an effect on mood as a reaction to diminished function.  This relationship would not be easy to untangle.  Qualitative data could shed light on how people with PD perceive depression. A qualitative study by Oehlberg found that people attribute depression to the diagnosis of Parkinson's, to the Parkinson's itself, reaction to deteriorating function, stigma, and other events in their life. This for some, the perception is that it is part of the condition itself, for others it is a complication, for others unrelated.  As there is a substantial literature that depression affects gait in people without motor effects of a diagnosis of Parkinson's we placed mood as a variable influencing gait parameters.

Belvederi Murri M, Triolo F, Coni A, et al. Instrumental assessment of balance and gait in depression: A systematic review. Psychiatry Res 2020;284:112687. (In eng). DOI: 10.1016/j.psychres.2019.112687   Oehlberg K, Barg FK, Brown GK, Taraborelli D, Stern MB, Weintraub D. Attitudes regarding the etiology and treatment of depression in Parkinson's disease: a qualitative study. J Geriatr Psychiatry Neurol. 2008 Jun;21(2):123-32. doi: 10.1177/0891988708316862. PMID: 18474721; PMCID: PMC2680384.

Reviewer 4 Report

Comments and Suggestions for Authors

·         The title is not clear and is not in line with the hypothesis. What is meant by the link in the title? Which outcome measure tests the brain health status in this study?

·         The literature review in the introduction section is not in line with the study hypothesis. What is the gap in the literature ? What is the rationale of the study ? What is the implication?

·         The sample size nis ot adequate enough for structural equation modelling

·         What is the design of the study? What are the eligibility criteria? How and from where were the participants recruited? All these details are required in the methodology

·         What is the reason for excluding participants with cognitive impairment? Cognitive impairment is common in PD. This will affect the generalizability of the study

·         What is the reliability and validity of the heel 2 toe sensor used in the study?

·         I have concerns regarding the scales used in the measurement of depression and fatigue. The visual analogue health scale is not a robust scale for this purpose

·   The concentration was assessed by one item in the PDQ scale. This is not a valid method for assessing concentration

·         There is no critical analysis of the results in the discussion. In addition, all the findings of the study have to be discussed in the discussion.

Author Response

Reviewer 4

The title is not clear and is not in line with the hypothesis. What is meant by the link in the title? Which outcome measure tests the brain health status in this study?

The title, "Evidence for the link between brain health states, kinematic gait parameters, and physical function in people with Parkinson’s Disease," reflects the study’s focus on exploring how non-motor symptoms, such as mood, fatigue, and concentration (proxies for brain health), influence gait and physical function in PD. Using path analysis, the study identifies direct and indirect relationships between these domains, highlighting their interconnected nature. The term "link" captures the study’s aim to bridge brain health, gait mechanics, and functional outcomes, making the title both accurate and aligned with its objectives.

The literature review in the introduction section is not in line with the study hypothesis. What is the gap in the literature ? What is the rationale of the study ? What is the implication?

Thank you for your valuable feedback. The introduction addresses the focus of rehabilitation on motor symptoms in PD, emphasizing the importance of walking and balance for daily functioning, as supported by systematic reviews [11]. It identifies a gap in the literature regarding how non-motor symptoms, such as depression, fatigue, and cognitive impairment, influence kinematic gait parameters and physical function [29-34]. Our study aims to fill this gap by exploring the direct and indirect effects of non-motor symptoms on gait mechanics and physical outcomes, which are currently under-studied [35,36].

The sample size nis ot adequate enough for structural equation modelling

Thank you for your feedback regarding the adequacy of the sample size for structural equation modeling (SEM).
Focus on Parsimony: We intentionally designed a parsimonious model, limiting the number of parameters and focusing on strong correlations between variables to minimize potential bias due to sample size constraints.
Effect Size Estimation: Using a sample size calculator for SEM, we determined that the minimum sample size required to detect an effect size of 0.5 or greater was 23 participants. Thus, while our sample size is adequate for estimating effect sizes, it is insufficient for testing the full model structure.
Caution in Interpretation: Given these limitations, we interpret our findings as exploratory and hypothesis-generating rather than confirmatory. These results provide preliminary insights into the relationships between non-motor symptoms, gait parameters, and physical function. The limitations of the sample size have been acknowledged and are discussed in the methodology and discussion sections, along with recommendations for future studies with larger sample sizes.We appreciate the opportunity to clarify these points and have ensured they are thoroughly addressed in the manuscript.

What is the design of the study? What are the eligibility criteria? How and from where were the participants recruited? All these details are required in the methodology

Thank you for your feedback. The methodology section provides detailed information about the study design, eligibility criteria, and participant recruitment process:
Study Design:
This study is a secondary analysis of baseline data from a randomized pilot and feasibility study. The aim of the trial was to evaluate the feasibility and acceptability of using the Heel2Toe™ sensor and estimate changes in gait patterns and walking capacity after training.
Eligibility Criteria:
Participants included individuals with Parkinson’s Disease (PD) who exhibited gait impairments but did not require a walking aid (Hoehn and Yahr scale 2-3). Exclusion criteria included a Montreal Cognitive Assessment (MOCA) score ≤ 25, indicating cognitive impairment.
Participant Recruitment:
Participants were recruited from the Movement Disorders Clinics at McGill University Health Center and the Quebec Parkinson Network. All participants provided written informed consent and were assessed during the "on" phase of their medication schedule.
These details are explicitly stated in the methodology section to ensure clarity and transparency about the study’s framework.

What is the reason for excluding participants with cognitive impairment? Cognitive impairment is common in PD. This will affect the generalizability of the study

Our ethics committe prohibits the recruitment of people with cognitive impairment into research projects unless they have a mandated carer.  Ethics notwithstanding, exclusion of participants with cognitive impairment ensures the validity and reliability of our findings by reducing confounding factors.

1. Nucleus basalis of Meynert degeneration precedes and predicts cognitive impairment in Parkinson’s disease
Schulz1, Pagano2, Bonfante3 et al. 2018Brain 2. Clinical variables and biomarkers in prediction of cognitive impairment in patients with newly diagnosed Parkinson's disease: a cohort study
Schrag1, Siddiqui2, Anastasiou3 et al. 2017The Lancet Neurology 3. Comparison of the Montreal Cognitive Assessment and Mini Mental State Examination Performance in Patients with Parkinson’s disease with w Low Educational Background
Martínez-Ramírez1, Rodríguez‐Violante2, Gonzàlez‐Latapi3 et al. 2014RNIJ 4. Methodological recommendations for cognition trials in bipolar disorder by the International Society for Bipolar Disorders Targeting Cognition Task Force
Miskowiak1, Burdick2, Martínez-Àran3 et al. 2017Bipolar Disorders

What is the reliability and validity of the heel 2 toe sensor used in the study?

The Heel2Toe™ sensor's reliability and validity are supported by its classification as a Class I medical device by Health Canada. Previous research demonstrated its accuracy in detecting proper heel strikes (94%) and its ability to provide effective real-time auditory feedback for gait improvement in people with Parkinson’s disease. Additionally, proof-of-concept studies have shown clinically meaningful improvements in gait parameters for participants using the device. All of the references to earlier work are cited in the trial from which the data was drawn.

Technical refinements addressed earlier usability challenges, ensuring the sensor's feasibility for independent use at home. These characteristics highlight the Heel2Toe™ sensor as a reliable and valid tool for gait analysis and intervention. The supporting evidence for the sensor's reliability and validity has been provided in the references listed as numbers 36–37 in the manuscript.

6. Vadnerkar A, Figueiredo S, Mayo NE, Kearney RE. Classification of gait quality for biofeedback to improve heel-to-toe gait. Annu Int Conf IEEE Eng Med Biol Soc 2014;2014:3626-9. DOI: 10.1109/EMBC.2014.6944408.
5. Vadnerkar A, Figueiredo S, Mayo NE, Kearney RE. Design and Validation of a Biofeedback Device to Improve Heel-to-Toe Gait in Seniors. IEEE J Biomed Health Inform 2018;22(1):140-146. DOI: 10.1109/JBHI.2017.2665519.
4. Mate KK, Abou-Sharkh A, Morais JA, Mayo NE. Real-Time Auditory Feedback-Induced Adaptation to Walking Among Seniors Using the Heel2Toe Sensor: Proof-of-Concept Study. JMIR Rehabil Assist Technol 2019;6(2):e13889. DOI: 10.2196/13889.
3 Carvalho LP, Mate KKV, Cinar E, Abou-Sharkh A, Lafontaine AL, Mayo NE. A new approach toward gait training in patients with Parkinson's Disease. Gait Posture 2020;81:14-20. DOI: 10.1016/j.gaitpost.2020.06.031.

I have concerns regarding the scales used in the measurement of depression and fatigue. The visual analogue health scale is not a robust scale for this purpose

This concern was raised by other reviewers and addressed above.  As the study focused on motor response, the questionnaire component was kept to a minimum.  In addtion, ref 43 demonstrates the relationship between single items and full measures of these constructs. Other research supports this choice.

1. Health anxiety and depression in patients with fibromyalgia syndrome
Uçar1, Sarp2, Karaaslan3 et al. 2015J Int Med Res  2.Dalfampridine effects on cognition, fatigue, and dexterity
Korsen1, Kunz2, Schminke3 et al. 2016Brain and Behavior 3.Validation of the Serbian version of inflammatory Rasch‐built overall disability scale in patients with chronic inflammatory demyelinating polyradiculoneuropathy
Perić1, Božović2, Pruppers3 et al. 2019J Peripheral Nervous Sys  4.Effect of Progressive Self-Focus Meditation on Attention, Anxiety, and Depression Scores
Leite1, Ornellas2, Amemiya3 et al. 2010Percept Mot Skills

· The concentration was assessed by one item in the PDQ scale. This is not a valid method for assessing concentration

This concern was raised by other reviewers and addressed above. 

 There is no critical analysis of the results in the discussion. In addition, all the findings of the study have to be discussed in the discussion.

Thank you for your feedback. We reviewed the discussion.  We feel that the discussion critically analyzes the study’s findings and addresses all major results: Key Findings:Fatigue and mood significantly influenced kinematic gait parameters and physical function. Fatigue affected heel strike, impacting gait stability, while mood influenced push-off and walking endurance, highlighting the importance of addressing non-motor symptoms to improve physical function.Indirect Effects: The discussion details the indirect effects of non-motor symptoms on physical function, mediated by gait parameters, such as fatigue’s influence on Neuro-QoL via heel strike and mood’s effect through push-off.Comparison to Literature:Findings are contextualized with prior research, reinforcing the clinical significance of the relationships observed and supporting the study’s conclusions. Strengths and Limitations: The study’s use of path analysis is highlighted as a strength, while limitations, including the small sample size and cross-sectional design, are acknowledged with suggestions for future longitudinal studies.

Round 2

Reviewer 3 Report

Comments and Suggestions for Authors

Dear authors,

thanks for considering my previous comments. I appreciate also the references that you provided with regards to depression and Parkinson's disease. I have no further comments.

Author Response

Comments

Dear authors,

thanks for considering my previous comments. I appreciate also the references that you provided with regards to depression and Parkinson's disease. I have no further comments.

Response 

Thank you for your kind feedback and for taking the time to review our manuscript. We appreciate your acknowledgment of the references provided and are glad to hear that you have no further comments. Your input has been invaluable in strengthening our work.

Best regards,

Reviewer 4 Report

Comments and Suggestions for Authors

The authors have addressed most of the queries. However, there few queries in the author's response that are not satisfactory 

1. VAS is being used for the measurement of depression and fatigue. This scale is not robust in measuring these outcome measures. Authors might have used scales such as the Beck Depression Inventory and Parkinsons' fatigue scale. This has to be addressed and added as a limitation of the study 

2.  I have concerns regarding the term 'health state' in the title. It is better to use the specific outcome measure names in the title 

Author Response

Comments 1

The reviewer pointed out that the Visual Analog Scale (VAS) is not robust for measuring depression and fatigue. They suggested using scales such as the Beck Depression Inventory or Parkinson's Fatigue Scale.

Response:
We appreciate the reviewer’s observation regarding the use of the VAS for measuring depression and fatigue.  We have added this as a limitation in the discussion section:
"A notable limitation of this study is the relatively small sample size. While the results provide valuable insights, the generalizability of our findings may be limited. We also had a limited portfolio of measures which was necessary to reduce respondent burden so the focus could be on the processes of the trial. Thus, our measure of cognition was restricted to a single item for concentration. The observation that concentration did not have any impact may be due to under-specification of this important construct. A richer set of measures for mood and fatigue would have been optimal, allowing for a structural equation model to be fit to the data and permit a focus on constructs rather than individual measures, which less accurately reflect the constructs of interest.”

Comment 2:
The reviewer expressed concerns about the use of the term "health state" in the title and suggested specifying the outcome measures instead.

Response:
We appreciate this constructive suggestion and have revised the title accordingly. The new title, "Evidence for the Link Between Non-Motor Symptoms, Kinematic Gait Parameters, and Physical Function in People with Parkinson’s Disease," provides greater clarity by explicitly naming the key outcome measures studied.
